# Assessment of Dietary and Lifestyle Quality among the Romanian Population in the Post-Pandemic Period

**DOI:** 10.3390/healthcare12101006

**Published:** 2024-05-14

**Authors:** Magdalena Mititelu, Violeta Popovici, Sorinel Marius Neacșu, Adina Magdalena Musuc, Ștefan Sebastian Busnatu, Eliza Oprea, Steluța Constanța Boroghină, Andreea Mihai, Costin Teodor Streba, Dumitru Lupuliasa, Emma Gheorghe, Nadin Kebbewar, Carmen Elena Lupu

**Affiliations:** 1Department of Clinical Laboratory and Food Safety, Faculty of Pharmacy, “Carol Davila” University of Medicine and Pharmacy, 020956 Bucharest, Romania; magdalena.mititelu@umfcd.ro (M.M.); nadin.kebbewar@mst.umfcd.ro (N.K.); 2Center for Mountain Economics, “Costin C. Kiriţescu” National Institute of Economic Research (INCE-CEMONT), Romanian Academy, 725700 Vatra-Dornei, Romania; violeta.popovici@ce-mont.ro; 3Department of Pharmaceutical Technology and Bio-Pharmacy, Faculty of Pharmacy, Carol Davila University of Medicine and Pharmacy, 020945 Bucharest, Romania; dumitru.lupuliasa@umfcd.ro; 4Institute of Physical Chemistry—Ilie Murgulescu, Romanian Academy, 060021 Bucharest, Romania; 5Department of Cardio-Thoracic Pathology, Faculty of Medicine, “Carol Davila” University of Medicine and Pharmacy, 050474 Bucharest, Romania; stefan.busnatu@umfcd.ro; 6Microbiology Department, Faculty of Biology, University of Bucharest, 060101 Bucharest, Romania; eliza.oprea@g.unibuc.ro; 7Department of Complementary Sciences, History of Medicine and Medical Culture, Carol Davila University of Medicine and Pharmacy, 050474 Bucharest, Romania; steluta.boroghina@umfcd.ro; 8Department of Pulmonology, University of Medicine and Pharmacy of Craiova, 200349 Craiova, Romania; andreea.mihai@umfcv.ro (A.M.); costin.streba@umfcv.ro (C.T.S.); 9Department of Preclinical Sciences I—Histology, Faculty of Medicine, “Ovidius” University of Constanta, 900470 Constanta, Romania; emma.gheorghe@365.univ-ovidius.ro; 10Department of Mathematics and Informatics, Faculty of Pharmacy, “Ovidius” University of Constanta, 900001 Constanta, Romania; clupu@univ-ovidius.ro

**Keywords:** healthy diet, balanced lifestyle, metabolic disorders, healthy habits, public health, nutritional education

## Abstract

Background: The pandemic caused by the SARS-CoV-2 virus demonstrated the importance of prevention through a healthy diet and lifestyle, the most vulnerable people being those with severe chronic conditions, those who are overweight, and those with an unbalanced immune system. This study aims to examine the nutritional status and lifestyle behaviors of the Romanian population. Methods: The evaluation of the eating habits and lifestyle of the Romanian population in the post-pandemic period was carried out based on a cross-sectional observational study with the help of a questionnaire. Results: A total of 4704 valid answers were registered (3136 female and 1568 male respondents). Among the respondents, most of them belong to the young population, 2892 between the ages of 18 and 40, i.e., 61.5%. Most male respondents are overweight (1400) and obese (780). Most respondents indicated a tendency to consume 1–2 meals per day irregularly (*p* = 0.617). Only 974 respondents adopted a healthy diet, and 578 a healthy lifestyle. Conclusions: The present study reports low adherence to a healthy diet (20.7%) and healthy lifestyle (12.28%), especially among the young population (<30 years). In the current context, it reports a reduced tendency to consume vegetables and fruits among the population, below the daily average recommended by the nutrition guidelines, a tendency towards sedentary behavior, and even deficient hydration of some of the respondents; these negative aspects can create a long-term series of nutritional and psycho-emotional imbalances. Our results evidence that complex surveys among the population are regularly required to investigate nutritional or lifestyle deficiencies; moreover, it could be helpful in further educational measures in nutrition, food, and environmental safety.

## 1. Introduction

The complex relationship between food, diet, and health lies at the heart of numerous public health challenges worldwide. Individuals’ dietary choices can profoundly influence their overall well-being, encompassing physical, mental, and emotional health dimensions. Additionally, the quality of food and lifestyle strongly affect the quality of life and the body’s resistance to the aggressions of environmental factors (pollution, microorganisms, climate changes). The statistics regarding the categories most vulnerable to the attacks of viruses indicate the elderly population, and people with severe chronic conditions (diabetes, cardiovascular diseases, lung diseases, oncological diseases, morbid obesity), triggered and primarily aggravated by significant nutritional imbalances [1,2,3]. The World Health Organization (WHO) indicates that obesity complications caused approximately 90% of deaths in 2021. The leading factors of the increase in the rate of non-communicable diseases are smoking, physical inactivity, excessive alcohol consumption, and unhealthy diet [3]. Many countries have begun to place increasing emphasis on stimulating nutritional programs for the prevention and balancing of nutritional imbalances. At the European Level, a series of cooperative multinational programs have been financed by stimulating collaborations between European Union states in order to reduce the incidence of obesity among the population, stimulating physical activity and strengthening healthy eating habits for the prevention of metabolic diseases, such as the European ERA4Health program [4].

The emergence of the COVID-19 pandemic has introduced unprecedented disruptions to global health. Beyond the immediate health threats posed by the novel virus, the pandemic has triggered widespread ramifications across healthcare systems, economies, and societal norms. Exploring the multifaceted repercussions of COVID-19 on health is essential for devising comprehensive strategies to mitigate its effects and enhance resilience in the face of future pandemics. On 5 May 2023, WHO declared that the threat posed by the COVID-19 pandemic has finally ended after more than three challenging years for global public health [5]. Throughout this period, the world faced immense social, economic, and health challenges, witnessing a significant loss of lives due to the SARS-CoV-2 virus, with a total death toll of 6,953,743 [6].

A healthy lifestyle means more than a set of rules to prevent or treat specific conditions; it represents a life concept that allows us to live healthily and build a world where people reach maximum physical and emotional potential [7]. A healthy lifestyle requires a constant balance between a harmonious diet, physical activity, adequate hydration of the body, rest, the absence of unhealthy habits (smoking, drug use, excessive alcohol consumption), as well as an environment as clean as possible, free of toxic agents and pathogens [8,9,10,11]. All these recommendations are ideal for increasing longevity, maintaining emotional well-being and mental balance, maintaining good general condition, and combating risk factors that endanger health. A healthy diet must ensure sufficient macro- and micronutrients to meet the body’s physiological needs, which differ from one organism to another, depending on sex, age, physiological state, physical activity, and metabolism [12,13,14]. For a balanced and correct diet, we must consume plenty of fruits and vegetables because they represent an essential source of antioxidants, vitamins, mineral salts and fibers, and macronutrients. It also involves consuming diverse foods from all groups, rich in valuable nutrients, for all the body’s needs [15,16,17,18]. Food is recommended to be consumed fresh or prepared using healthy cooking techniques. Vegetables and fruits should be eaten raw as much as possible to preserve their content rich in enzymes, vitamins, and minerals [19,20,21,22,23,24,25]. Food sources must be controlled, thus avoiding contamination with microplastics, pesticides, or heavy metals. Polluted marine areas for harvesting seafood or fish should also be avoided [26,27,28]. Eating contaminated food can seriously affect health [29,30,31]. In addition, it is essential to avoid excessive administration of drugs or their consumption without the supervision of a specialist [32,33] due to their potential toxicity. 

Physical activity and sports practice are permanent means of maintaining health, increasing work capacity, and prolonging life, so it is recommended that people exercise for at least 30 min every day [34,35,36,37,38,39,40]. Under normal conditions, adults metabolize 2.5–3 L of water daily, covered by water consumption as such and the water contained in the ingested food. Water intake and loss are regulated by thirst and hormonal mechanisms. During intense physical exercises, the need for water is increased, as it also happens in the case of increased temperatures in the external environment [41,42,43,44,45,46]. On the other hand, sleep is fundamental to the individual’s physical and psycho-emotional health; 7–9 h of sleep per night are recommended for people between 18 and 64 years old and 7–8 h for people over 65 years old [47,48,49,50].

An analysis of the factors involved in the life span highlights an involvement percentage of approximately 25% of the genetic influence, while the environment and the lifestyle exert 75%. As a result, lifestyle and eating habits are essential for life quality and longevity [51,52,53,54], simultaneously influenced by the emotional state (Figure 1).

According to the data published in 2020 related to Romania’s health profile, life expectancy, which increased after 2000, was seriously affected by the COVID-19 pandemic, falling far below the European Union (EU) average. In addition, the rate of alcohol consumption and unhealthy food among Romanians is above the EU average. Generally, behavioral risk factors (tobacco consumption, alcohol, unhealthy diet, and lack of physical activity) and environmental ones (pollution) cause over half of the number of deaths, and the mortality rate from treatable causes in Romania is the highest in the EU [55].

In this context, the present study aims to evaluate the nutritional status and lifestyle among the Romanian population, reporting the results to the standards of balanced nutrition by calculating adherence to a healthy diet and lifestyle. Behavioral risk factors (diet, lifestyle) have an important impact on the quality of life. Therefore, it is essential to make a periodic evaluation of them by referring to the standards of normality in the nutrition and healthy lifestyle guides [56] to be able to formulate effective strategies for mitigating the negative impact in case of deviations from normality. In this work, an investigation of the dietary patterns and lifestyle of the post-pandemic Romanian people was performed using a questionnaire to explore connections between dietary choices, behavioral aspects, and individuals’ health status. This survey evaluated how respondents’ commitment to a healthy lifestyle and well-rounded diet correlated with their overall health. An extensive statistical analysis highlighted the deviations from the healthy lifestyle. Our results could be a potential database for formulating suitable solutions to help the population move closer to normal standards.

## 2. Materials and Methods

### 2.1. Study Design

A cross-sectional observational study was conducted using an online questionnaire via the Google Forms platform from 15 May to 15 August 2023, to assess the Romanian population’s dietary patterns and lifestyle choices. The survey, comprising 48 items, was circulated through official student networks, professional organizations, and social media platforms in Romania. Various Romanian institutions were involved, and the study was conducted on a sample of the country’s general population. They agreed to disseminate the questionnaire to employees or students through the institutional email, and the responses of the volunteer participants after completing the questionnaire were automatically recorded on the Google Forms platform in real time. Respondents were also asked to share the survey link with their colleagues and friends. The questionnaire aims to collect some socio-demographic information (age, sex, occupation, area of residence, studies), anthropometric data (height, weight), information related to eating habits correlated with lifestyle, psycho-emotional state correlated with health problems, as well as the identification of factors that affect the psycho-emotional balance of the respondents. The final database was downloaded as a Microsoft Excel spreadsheet. Participation in the survey was voluntary and open to individuals over 18 years old residing in Romania. Participants were required to provide individual consent, with assurances of identity protection. In the first part of the questionnaire, the respondents were informed about the study’s purpose and coordinating team.

Moreover, participants were guaranteed confidentiality concerning their sensitive personal information in compliance with the General Data Protection Regulation (GDPR). This assurance enabled the utilization and publication of the research results. The questionnaire dissemination process implied no discrimination based on gender, religion, or political beliefs. The present study agrees with the updated Declaration of Helsinki and received approval from the University of Medicine and Pharmacy Craiova Ethics Committee (No. 121/7 May 2023). 

### 2.2. Questionnaire Validation

Initially, the questionnaire underwent a pilot phase and was tested among a cohort of 250 participants over 18 years. Subsequently, a panel of 3 experts was convened to scrutinize the responses collected during the pilot phase, aiming to validate and enhance the questionnaire. The content validity ratio (CVR) and content validity index (CVI) were calculated [57,58]. Connelly (2008) proposed that the pilot study sample size should constitute a minimum of 10% of the intended sample size for the overall study [59]. In adherence to this recommendation, this study utilized the sample size calculation formula outlined by Viechtbauer et al. (2004), with parameters set at a 95% confidence level and a margin of error of ±4% [60]. The calculated sample size required for the pilot study was 94, which was deemed necessary for detecting feasibility issues. Given the potential variability in eating habits across different geographical regions of Romania, 250 participants were recruited.

In the pilot study, exploratory factor analysis (EFA) was conducted. The Kaiser–Meyer–Olkin (KMO) statistic yielded a value of 0.827, indicating the adequacy of the data for EFA [61,62]. Additionally, Bartlett’s Test of Sphericity returned a statistically significant result (*p* < 0.001), suggesting the absence of multicollinearity within the dataset [63].

The 3 experts processed the final form of the questionnaire in such a way as to increase its clarity and accuracy, and the final corrected version after irrelevant items were removed is presented as Appendix A. The internal consistency of the questionnaire was analyzed using the Cronbach’s α coefficient. For the present questionnaire, the value of Cronbach’s α was 0.84, indicating good internal consistency and reliability of the scale [64,65].

### 2.3. Statistical Analysis

All variables are displayed using absolute frequencies (n) and relative frequencies (%). Recognizing that age significantly influences eating habits and lifestyle, shaped by various experiences, environmental shifts, and potential health issues, we categorized respondents into five age groups for data analysis. Hence, we restructured the continuous age variable into five categories: young individuals up to 30, early adults aged 31 to 40, middle-aged adults between 41 and 50, aging adults between 51 and 60, and seniors over 60. Additionally, we classified the people using Body Mass Index (BMI) value into four groups: underweight, normal weight, overweight, and obese.

The data were completed with new categorical variables named “Adherence to a healthy diet”, which was constructed based on the answers to questions 9–21, and “Adherence to a healthy lifestyle” based on the answers to questions 27–28, 36, 38, 39, 43, and 44, all questions having answers from 1 (“very little” or “not at all”) up to 5 (“a lot” or “always”). 

In assessing “Adherence to a healthy diet”, the summation of responses constituted a raw score, subsequently converted into a T score (standardized). This conversion utilized an average of 54 and a standard deviation of 7 (ranging from a minimum of 28 to a maximum of 73). Scores surpassing 60 signified maximum adherence to a healthy diet, while those falling below 42 indicated adherence to an unhealthy diet. 

In evaluating “Adherence to a healthy lifestyle”, the aggregation of responses generated a raw score converted into a T score (standardized). This transformation utilized an average of 20 and a standard deviation of 4.5 (ranging from a minimum of 8 to a maximum of 33). Scores exceeding 26 indicated the highest adherence to a healthy lifestyle, while scores below 17 indicated an unhealthy one. 

Pearson’s Chi-square and Fischer tests were used to examine the potential association among variables (Gender, Body Mass Index groups, Residence, Education, Marital status, Employment status, Adherence to a healthy diet, and lifestyle) in connection with other ones included in the study, and Spearman’s correlation coefficient was calculated for associated continuous variables. All statistical analysis and graphic presentations were performed in IBM SPSS (Version 23.0) [66] and XLSTAT software (https://www.xlstat.com/en/download/customer/xlstat, accessed on 14 June 2023) [67]; the statistical significance level was considered at *p*-value < 0.05.

Finally, we analyzed the association between the outcome variables “Adherence to a healthy diet” and “Adherence to a healthy lifestyle” with the predictor variables such as Gender, Age groups, and BMI groups by applying logistic regression for a multinomial model. The results were expressed by odds ratio (OR), confidence interval (95%CI), and the associated *p*-values [68].

## 3. Results

### 3.1. Socio-Demographic and Anthropometric Data

After the questionnaire’s distribution, 4704 valid responses (66.7% from female respondents and 33.3% from male ones) were collected, with a confidence interval of 95% and a margin of error of ±3.05%. Utilizing anthropometric measurements (weight and height) to compute Body Mass Index (BMI) via the Quetelet equation [body mass (kg)/height (m^2^)], the data were interpreted following the WHO’s criteria [69]. Upon analyzing the anthropometric data, it was noted that most respondents, especially women (1726), fell into the normal weight category (2288). Conversely, a considerable proportion of the male group, totaling 718, was overweight [70]. Regarding their age, 61.5% are young, up to 40. All socio-demographic and anthropometric characteristics of the participants are presented in Table 1.

As can be seen from the data presented in Table 1, most participants in the study had higher education (61.6%), 17.8% were students, and 67.5% were employed. Most respondents live in the city (77.6%) and are married (57.6%).

### 3.2. Adherence to a Healthy Diet

In determining the respondents’ adherence to a healthy diet, the assessment relied on the principles of a well-rounded and nutritious diet, as outlined in the introductory section. Healthy food intake with heightened nutritional value is aligned with the nutritional guidelines, emphasizing both frequency and quantity of consumption. An unhealthy diet, accompanied by unfavorable habits, corresponded to the lowest score, while a balanced diet with healthy habits received the highest score. A moderately healthy diet fell under the middle category. As depicted in Table 2, most respondents were categorized under the moderately healthy diet group (3476), with 20.7% falling under the healthy diet category and 5.4% under the unhealthy diet category. Notably, a significant proportion of individuals in the unhealthy diet group were under 40 years old (73.2%), classified as overweight or obese (58.2%), residing in urban areas (75.6%), and engaging in daily work-related movements (48.8%). Regarding adherence to a healthy diet within specific age groups, 16.62% of individuals up to 30 years old, 23.76% up to 40 years old, 21.47% up to 50 years old, and 22.45% up to 60 years old adhered to a healthy diet. Interestingly, 34.15% of individuals over 60 who completed the questionnaire were in the healthy diet group. Predominantly, respondents in the healthy diet group were female (698), had a normal weight (524), lived in urban areas (788), possessed higher education (66.9%), were married (596), and commuted daily for work (486).

According to the probability associated with the Chi-square tests, the variables that most influence the choice of group are Age: χ^2^ = 51.082, *p* < 0.0001, Gender: χ^2^ = 9.763, *p*= 0.0076, BMI_group: χ^2^ = 22.368, *p* = 0.001, Level of education: χ^2^ = 15.96, *p* = 0.043, and Marital status: χ^2^ = 9.855.96, *p* = 0.0429.

The healthy diet group was considered as a reference in the multinomial logistic regression model (Figure 2), and in the model, age, gender, BMI, Level of education, and Marital status appeared to be significant predictors of leading an unhealthy diet (*p* < 0.05). 

Men were 1.93 times more likely to achieve an unhealthy diet (95%CI = 1.26–2.96, *p* = 0.024). Individuals between 18 and 30 years were more likely to score 1 and 2 for unhealthy diets (OR = 3.82, 95%CI = 1.76–4.28, *p* = 0007) than those aged 51–60, used as the reference category. Single respondents were likelier to have an unhealthy diet than their married counterparts (OR = 1.89, 95%CI = 1.26–2.85, *p* = 0.02). Moreover, participants with postsecondary education were more inclined to exhibit an unhealthy diet compared to those with higher education as the reference category (OR = 2.43, 95%CI = 1.23–4.8, *p* = 0.01).

Obese participants demonstrated a higher likelihood of having an unhealthy diet (OR = 1.88, 95%CI = 1.07–3.30), whereas those with normal weight were less likely to have an unhealthy diet (OR = 0.5, 95%CI = 0.3–0.82, *p* = 0.028).

In the multinomial logistic regression analysis, several significant factors were linked to maintaining a moderately healthy diet, including age, gender, and educational level. Those aged over 60 years were less likely to uphold a moderately healthy diet (OR = 0.43, 95%CI = 0.26–0.7). Men exhibited a 1.27 times higher likelihood than women to achieve a moderately healthy diet (OR = 1.27, 95%CI = 1.08–1.61, *p* = 0.085). Moreover, individuals with a general/primary education level (OR = 1.92, 95%CI = 1.1–3.34, *p* = 0.02) were more inclined to maintain a moderately healthy diet than those with higher educational attainment.

Young people (18–30 years) showed the lowest adherence to a healthy diet among all the age groups participating in the study. As seen from Figure 3, in the diet of this age category, foods rich in fats and carbohydrates (fats, eggs, bread, sweetened carbonated drinks) predominate, while the consumption of vegetables, fruits, fish, and seafood is considerably reduced.

Using correspondence analysis (CA), significant differences (χ^2^ = 28.56, *p* < 0.0001) among four BMI groups and the type of food consumed were identified (Figure 4); namely, low consumption of vegetables, fruits, fish, seafood, cereals, and dairy products and more significant amounts of food, fats, sweets, and pastries in obese and underweight respondents compared to those with normal BMI. 

With CA, significant differences (χ^2^ = 27.43, *p* = 0.0002) were identified among the four age groups (Figure 5). The bi-plot indicated that 99.06% of the variability observed could be attributed to the two principal components (F1: 85.35%, F2: 13.71%). The younger generations aged 18–30 preferred sweets, pastries, cereals, and pasta, while those aged 41–50 had vegetables, fruits, and meat. Respondents over 50 had a significantly lower consumption of fish, seafood, and high-fat products compared to the other age groups.

The breakdown of BMI and gender concerning adherence to a healthy diet (Figure 6) reveals that individuals, regardless of gender, predominantly within the normal weight or underweight categories, align themselves with a healthy or moderately healthy diet. Conversely, those classified as overweight or obese are more inclined to follow a moderately healthy or unhealthy diet (*p* ˂ 0.003).

### 3.3. Eating Habits and Lifestyle

The predominance of healthy habits (reduced alcohol consumption, smoking frequency, sports, recreational activities, sleep duration) and how meals are taken and distributed were examined to analyze the adherence to a healthy lifestyle.

According to the data presented in Table 3, 12.28% of the respondents belong to the group of those with a healthy lifestyle, 66.45% belong to the group of those with a moderately healthy lifestyle, and 21.25% belong to the group of those with an unhealthy lifestyle. Most of the respondents with increased adherence to an unhealthy lifestyle are young people up to 40 years old (65.6%), women (71%), overweight or obese (56.8%), they live in the urban environment (78%), are married (53.4%), and commute daily to work (50%). Respondents over 50 are more likely to be among those with increased adherence to a healthy lifestyle compared to those younger than them.

According to the probability associated with the Chi-square tests, the variables that most influence the choice of group are Gender: χ^2^ = 16.763, *p* = 0.0002, BMI group: χ^2^ = 81.752, *p* < 0.0001, Marital status: χ^2^ = 9.595, *p* = 0.0478, and Employment status χ^2^ = 34.644, *p* = 0.0017.

The healthy lifestyle group was considered the reference in the multinomial logistic regression model (Figure 7).

The multinomial logistic regression analysis found that statistically significant factors associated with leading an unhealthy and moderately healthy lifestyle were gender, BMI, Marital status, and Employment status. 

Men were 0.49 times less likely to achieve an unhealthy lifestyle (95%CI = 0.34–0.69, *p* < 0.0001) and 0.67 times less likely to achieve a medium unhealthy lifestyle (95%CI = 0.5–0.9, *p* = 0.0083) compared to women.

The obese people were more likely to have an unhealthy lifestyle (OR = 1.98, 95%CI = 1.17–3.37, *p* = 0.01), and those with normal weight and underweight were less exposed (OR = 0.32, 95%CI = 0.22–0.46, *p* < 0.0001; OR = 0.17, 95%CI = 0.08–0.36, *p* < 0.0001) and moderately healthy lifestyle (OR = 0.39, 95%CI = 0.22–0.46; OR = 0.57, 95%CI = 0.41–0.78).

Single respondents were more likely to have unhealthy lifestyles than married ones as reference category (OR = 1.66, 95%CI = 1.1–2.51, *p* = 0.0154). 

The participants who work in a mixed regime (OR = 0.42, 95%CI = 0.22–0.78, *p* = 0.0064) and teleworking (OR = 0.59, 95%CI = 0.35–0.99, *p* = 0.0493) were less likely to have an unhealthy lifestyle. The same observation regarding moderately unhealthy lifestyles is available for retired participants (OR = 0.46, 95%CI = 0.23–0.93, *p* = 0.031).

As can be seen from Figure 8, in the case of both sexes, the overweight and obese respondents are mainly found in the group of those with lower adherence to a healthy lifestyle.

Based on the responses gathered (Figure 9), a more significant inclination towards insomnia was evident among obese respondents, with 12% experiencing this issue, compared to other categories, notably contrasting with underweight individuals at 3%. Furthermore, within the obese category, a higher tendency to sleep less than 7 h per night was observed (43%) compared to other categories, particularly when compared to underweight respondents at 29%. Conversely, the underweight group displayed the highest tendency to sleep for 7–9 h per night (61%) in contrast to other categories, especially the obese group, where only 41% reported this sleep duration (*p* ˂ 0.001).

There is a close correlation between the quality of the diet and the adopted lifestyle, in the sense that people with greater adherence to a healthy diet and healthy lifestyle have optimal body weight; the lower the adherence, the greater the risk of imbalances related to problems that, in time, lead to health complications (Figure 10).

Concerning adopting a restrictive diet for weight reduction (Figure 11), 53.5% of respondents claimed they never employed such a regimen, 26.9% stated occasional use, and 19.6% acknowledged frequent use. Surprisingly, among those using restrictive diets were individuals classified as underweight, suggesting either excessively stringent diets causing bodily issues or a lack of awareness regarding their below-normal body weight. Notably, 23% of overweight and 38% of obese respondents frequently resorted to restrictive diets, while only 14% of individuals with normal weight reported frequent use.

Among those who never employed a restrictive diet, approximately 87% were underweight, 61% had normal weight, 49% were overweight, and 28% were obese (*p* ˂ 0.001). Female respondents predominantly admitted frequent use of restrictive diets (25%), contrasting sharply with male respondents at 8%. Additionally, 71% of male respondents stated they never opted for a particular diet for weight loss (*p* ˂ 0.001).

In line with the gathered responses, a significant portion of individuals with excessive weight—37% among the overweight and 53% among the obese—admitted to excessive food consumption (*p* ˂ 0.002). Interestingly, only 35% of underweight ones acknowledged insufficient food intake, suggesting a lack of awareness among respondents regarding their dietary habits corresponding to their body’s requirements. Notably, other underweight individuals did not acknowledge deficient food intake, although their body weight indicates otherwise. A mere 17.5% of respondents monitor their body weight, and a minimal part (1–2% across each BMI group) claimed to adhere to a nutritionist’s recommended food intake ratio.

The data revealed a similar distribution across categories regarding the periodic health status assessment. Approximately 28% of respondents never evaluate their health status, with 24.3% doing so very rarely. Only 34.5% habitually assess their health status annually, and merely 11.4% do so at least twice a year—likely individuals with significant health concerns. 

Analysis of the questionnaire data highlights that 47.2% of all respondents consume roughly 2 L of water daily. However, a notable proportion do not hydrate adequately—12.8% consume less than 1 L per day, and 33.6% consume approximately 1 L daily. Surprisingly, this pattern remains consistent across BMI groups despite the expectation that water intake should correlate with body mass, not solely physical activity. Specifically, 11% of overweight and 12% of obese individuals consume less than 1 L daily, while 33% of overweight and 30% of obese individuals consume around 1 L per day (*p* = 0.005).

The percentage of individuals engaging in daily sports activities for less than an hour or at least an hour is minimal, falling below 20%. Sedentary behavior is frequent among obese individuals (36% rarely engage in sports) and overweight ones (24% do not participate in sports, and 45% rarely do so). Surprisingly, even underweight respondents exhibit low activity levels (24% do not participate in sports, and 53% rarely do). Comparatively, individuals with normal weight show slightly better engagement in physical or sports activities than other BMI groups (*p ˂* 0.002). Although, as shown by the processed data from the questionnaire, very few respondents are used to frequently monitoring their body weight or calling a nutritionist to monitor their caloric intake from food; 37.7% of the respondents believe that they need the advice of a specialist to be able to choose healthy foods and 26.6% of them to eat balanced (Figure 12).

The leading causes that depreciate the state of health are considered stress (71.7%), lack of movement (55.1%), low sleep quality (48.5%), unhealthy diet (45.3%), and excessive work (34.1%). Only 29% of the respondents know the seriousness of pollution on their health (Figure 13).

Generally, the psycho-emotional state is mainly affected by stress (78.4%) and fatigue (58.1%). It is also negatively influenced by overworking, lack of communication, financial and family problems, and lifestyle (Figure 14). 

Figure 15 shows that the main problems affecting well-being are fatigue (52.5%) and nervousness (31%).

There is a relatively high percentage of respondents who admit that they do not know if they have health problems (almost 20%), which is to be expected considering that many of them either do not turn to specialized medical personnel when encountering health problems by calling the first phase of self-medication, or do not regularly access specialized medical personnel when they encounter problems (Figure 16).

The main factors blamed by the respondents for depreciating the lifestyle are sedentariness (51.8%), sleep quality (47.7%), and lack of free time (44.9%). For the adverse effects, a series of unsolved stressful problems, insufficient financial resources, and the quality of the food products consumed can be noted (Figure 17).

Spending free time is essential in restoring the body and balancing psycho-emotional health. The top preferences include watching movies and TV programs (62.2%) and activities with family or friends (61.8%). Also, the tendency to spend free time on social networks, in the open air, or reading as a recreation is noticeable (Figure 18).

## 4. Discussion

The processing of the socio-demographic data collected from the survey participants highlights the presence in particular of young people under the age of 50 (Table 1), who represent 83.5% of the total number of respondents, the majority coming from the urban area (77.6%) and having higher education (61.6%). More than 50% of the people interviewed commute to work every day and are married. The processing of the anthropometric data (Table 1) indicates that almost half of the respondents have problems with body weight (46.1%) because, according to BMI, they belong to the group of underweight (4.3%), overweight (28.3%), or obese (13.5%) people, which suggests from the start that there are nutritional imbalances. Regarding eating habits, the following aspects were observed: 30.3% of the respondents mainly consume saturated fats (butter in particular, but also lard or margarine), only 31.6% are used to consuming virgin or extra–virgin vegetable oils, approximately 60% consume vegetables and fruits very rarely or at most only one portion per day, 43.3% consume fish and seafood very rarely or not at all and 41.8% only once a week, 33% consume meat daily, 42% consume meat more than two portions per week, 44% do not hydrate appropriately by consuming up to 1 L of water per day. A first aspect worthy of note is represented by the fact that many of the respondents do not consume enough fiber. In combination with inadequate hydration of the body, there is an increased risk of accumulating toxins that can affect the state of health in the long term. The increased consumption of meat, correlated with a reduced intake of fiber and water, also represents a long-term aggravating factor that can lead to the appearance of metabolic diseases. Some studies indicate the tendency to reduce consumption of fruits and vegetables, especially among young people [71,72]. In a statistic published by the European Commission in 2019 based on a survey in the member states, there is an increased tendency to consume vegetables and fruits (at least 5 daily portions), especially among people with a high level of education. There are European countries where the population is used to consuming more significant amounts of vegetables daily (Spain, Portugal). However, Romania is included among the countries with a low daily consumption of vegetables [73]. In addition, surveys carried out during the pandemic to assess eating habits have indicated the reduced consumption of vegetable products among the Romanian population [74,75]. In order to increase the consumption of vegetables and fruits, it is necessary to implement some nutritional education actions starting from kindergartens and schools; moreover, paying particular attention to the menus in canteens in schools and kindergartens to include as many seasonal vegetables and fruits as possible. It is also necessary to carry out campaigns to raise awareness of the optimal hydration of the population, primarily since the state of dehydration of the body, even in reduced forms, affects the ability to concentrate and memory. Drivers who must react as efficiently as possible must know that inadequate long-term hydration can affect their driving skills and even cause accidents [76,77].

The low adherence of the respondents to a healthy diet (4.3%), according to the results obtained from the processing of the data collected with the help of the survey (Table 2), confirms the need to improve the eating habits of the Romanian population by stimulating the consumption of nutritious foods: vegetables, fruits, fish, seafood, unrefined vegetable fats. In addition to campaigns to promote nutritional information related to the quality of different food groups and their nutritional value, it would also be helpful to develop social measures aimed at stimulating domestic production of some essential foods (milk, eggs, cereals, vegetables, and fruits, fish) and to facilitate their commercialization on the domestic market by granting subsidies and facilities on the condition of limiting price increases in order to stimulate domestic consumption at all social levels of the population. It is known that shortening the distribution chain for food products usually implies the use of much smaller amounts of chemical additives.

A positive aspect worth noting is that in culinary preferences, food cooked at home, boiled, or steamed in the oven predominates. The preferred type of meat is represented by chicken (53.4%) and pork (30.5%).

In terms of lifestyle, the following aspects are noteworthy: most respondents have an irregular, even unbalanced, distributed meal schedule if we take into account the fact that 43.1% of them consume 1–2 meals a day, 40.8% of the respondents declared that they chaotically consume excess or insufficient food, 61.2% serve the meal in a hurry or during the meal they tend to do something else, 64.4% do not tend to do sports or exercise at all or only very rarely in the conditions in which 57.6% of the respondents work at the office or in front of the computer, and 27.3% spend more than 8 h a day in front of the computer, 18.5% spend around 6–7 h a day in front of the computer, and 22.3% spend about 4–5 h a day. Under these conditions, it is understandable that 40.4% of the respondents have insomnia or insufficient rest, 18.8% consider that they have a weakened immune system, 26.1% regularly use various methods to strengthen the immune system, 78.4% of the respondents are stressed, 52.5% are tired, and 31% are frequently tired. Alteration of the psycho-emotional status is the result of an unbalanced lifestyle, with a chaotic diet and low consumption of vegetable products, insufficient hydration, an accentuated tendency to be sedentary, a recreational activity that takes place predominantly in front of the television or on social networks (Figure 18), as well as a neglect of the periodic evaluation of the state of health. The neglect of the state of health is also noticeable by the reduced tendency of underweight or overweight people to resort to measures to help them balance the body, given that many respondents are aware of the negative aspects that affect their health: monitoring food consumption and body weight, hydration corresponding to the body (a measure that helps overweight people in particular to consume smaller amounts of food), calling on the services of a nutritionist. Furthermore, in this direction, a series of measures are required to help reduce imbalances: the promotion of outdoor party campaigns both through festive activities and through sports competitions aimed at the population, the intensification of recreational activities through exercise in schools and kindergartens in order to combat sedentary lifestyle, the implementation of mandatory legislative measures to monitor the state of health based on the package of free analyses existing in the rights of insured persons (mostly neglected rights), the introduction of psychological counseling services in large communities (schools, high schools, universities, large companies with over 21 employees). Excessive work, inadequate rest, lack of engaging in recreational activities that increase self-esteem (participation in socio-cultural activities), communication problems related to unhealthy habits, and an unbalanced diet not only affect the state of health but also produce disorders of psycho-emotional behavior such as states of nervousness, anxiety, depression, loss of appetite or emotional eating, altering the ability to work but also to integrate into communities (Figure 15 and Figure 17).

The positive aspects of the lifestyle highlighted by the present study consist of reduced consumption of alcoholic beverages among the respondents (63% declared that they consume very rarely or not at all) and diminished tendency to smoke. Comparing this rate with the results collected in the last 16 years [78] (70.4% are non-smokers), only 17.6% of the respondents declared that they smoke excessively.

Moreover, all these observed imbalances confirm the health problems highlighted by the National Study on the Prevalence of Diabetes, Prediabetes, Overweight, Obesity, Dyslipidemia, Hyperuricemia and Chronic Kidney Disease (PREDATORR), started in 2013 and carried out by the Romanian Society of Nephrology in partnership with The Romanian Society of Diabetes, Nutrition and Metabolic Diseases, for almost a year and a half, on a representative sample of approximately 3000 people (20–79 years), from 101 centers in Romania.

According to the PREDATORR epidemiological study, approximately 11.6% of the adult population of Romania (20–79 years old) has diabetes. This value is 31.8% higher than the global average (8.8%) reported and 70% above the European average. Following the study, it was also found that over 80% of the adult population of Romania aged between 20 and 79 years have abnormalities of serum lipids (19% have dyslipidemia, 81% have dyslipidemia with disorders of one or more lipid factors), while over 16% of Romanians in the same age range have hyperuricemia. Regarding body weight, the study data indicate that 31.4% of Romanians have first-degree obesity, 21.5% second-degree, and 2.7% have morbid obesity. Also, 34.6% are overweight and on the verge of obesity. The study also indicates a high prevalence of 6.7–7.7% (depending on the calculation formula) for chronic kidney disease and 61.7% for blood pressure. The study also highlighted the fact that about 2.4% of patients with diabetes were undiagnosed [79,80,81].

The implementation of measures to increase the quality of public health by promoting a favorable nutritional status and a healthy lifestyle correlated with an environment with as little pollution as possible is the key to ensuring an efficient workforce, a population with a low incidence of various diseases, including the mental nature with direct benefits on the public health system by reducing the costs related to the protection programs of the various chronic patients but also on the increase in the life expectancy of the population.

It is also necessary to boost the consumption of vegetable proteins (especially legumes, nuts, and seeds) in the context in which health specialists draw attention to the fact that animal breeding activity is becoming more and more polluting with the demographic expansion of the population at the world level. The aim is to create a sustainable food future that will serve more than nine billion people by 2050 [82].

In a cross-sectional study from 2023 conducted through an online survey with 1451 Saudi adults residing in Riyadh, Saudi Arabia, in the post-pandemic period, it was found that although there were respondents who improved the quality of their diet by reducing the consumption of unhealthy food, especially junk food and the increase in the consumption of vegetable products (28.9%), many respondents reported an increase in weight (40.9%) and 33% an increase in the consumption of junk food products, especially male respondents. The study draws attention to the fact that the signals related to the increase in the consumption of unhealthy food and its consequences manifested by weight gain and the impairment of well-being are worrisome [83].

Another study carried out among the population of South Africa, also based on a questionnaire, in which 4786 respondents participated, highlighted a worsening of food insecurity throughout the pandemic period, which led to significant nutritional imbalances among the population [84].

According to official reports from the USA, food insecurity among the population increased during the pandemic, but through the measures implemented by the American government, it was reduced in the post-pandemic period [85,86].

### 4.1. Limitation Section

Our study has several limitations, including the lower participation of male respondents, elderly participants, and rural residents. The obtained data report an alteration of the psycho-emotional state caused by a disordered lifestyle and unbalanced diet. More detailed studies are required to evaluate the degree of alteration of the psycho-emotional components under the influence of lifestyle and eating habits. Another limitation of this study is the reliance on self-reported BMI data provided by participants.

### 4.2. Practical Applications

The findings of this study have several implications for public health interventions, clinical practice, and policy development. The identification of high prevalence rates of unhealthy dietary habits and sedentary behaviors underscores the importance of targeted health promotion programs aimed at promoting healthier lifestyles among the Romanian population. These programs could include educational campaigns, community-based initiatives, and policy interventions aimed at promoting healthy eating habits, increasing physical activity levels, and reducing sedentary behaviors. Healthcare professionals can utilize the findings of this study to inform clinical practice and provide tailored interventions to patients at risk of developing chronic diseases associated with unhealthy lifestyle behaviors. By assessing patients’ dietary habits, physical activity levels, and other lifestyle factors, clinicians can develop personalized treatment plans aimed at improving health outcomes and reducing disease risk. Policymakers can leverage the evidence provided by this study to inform the development of policies and strategies aimed at addressing the social determinants of health and promoting population-wide health improvements. This may include initiatives to improve access to healthy food options, create supportive environments for physical activity, and implement regulations to reduce exposure to unhealthy food environments. The findings of this study can guide future research priorities in the field of public health and preventive medicine. Areas for further investigation may include exploring the underlying determinants of unhealthy lifestyle behaviors, evaluating the effectiveness of different intervention strategies, and assessing the long-term impacts of lifestyle interventions on population health outcomes.

## 5. Conclusions

The present study, performed on 4704 Romanian participants over the age of 18, reports low adherence to a healthy diet (20.7%) and healthy lifestyle (12.28%), especially among the young population (<30 years). In the current context, it highlights a series of problems related to the lifestyle and diet of the Romanian population. Our results evidence that complex surveys among the population are regularly required to investigate nutritional or lifestyle deficiencies. This study could be helpful in further educational measures in nutrition, food, and environmental safety. Moreover, data analysis could underline further administrative and legislative regulations regarding potential assistance in large autochthonous communities to increase the population’s adherence to a healthy diet by periodically evaluating their health status. It also highlights the progressive need for people’s involvement in environmental protection to ensure pollution reduction, a suitable lifestyle, and optimal psycho-emotional balance with beneficial effects on the well-being and health of the Romanian population.

## Figures and Tables

**Figure 1 healthcare-12-01006-f001:**
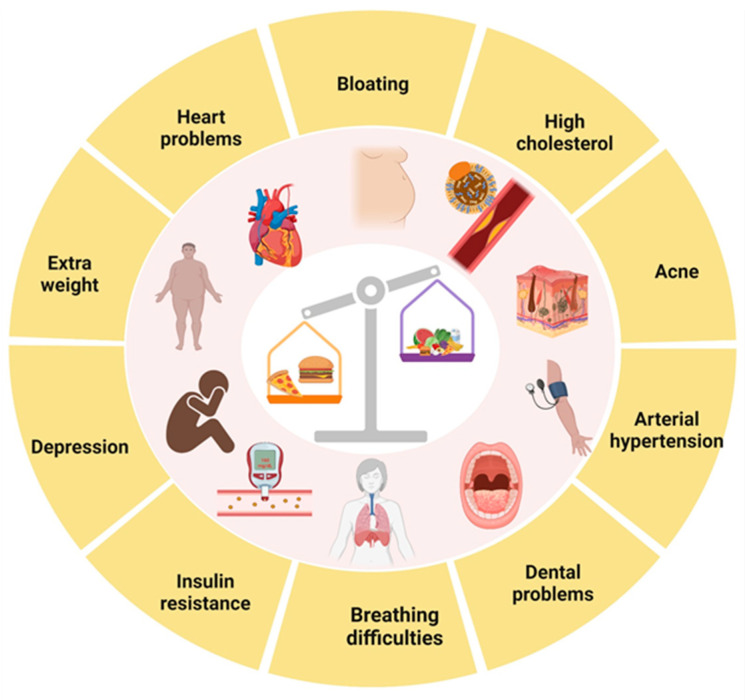
The influence of food quality on health. Created with BioRender (accessed on 25 November 2023).

**Figure 2 healthcare-12-01006-f002:**
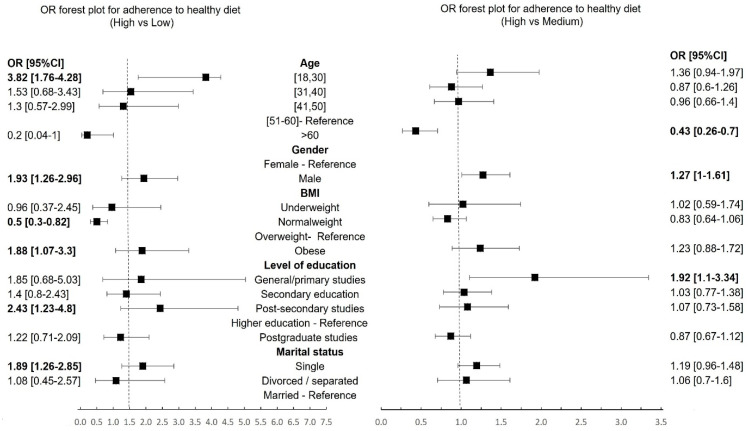
The multinomial logistic regression results with categories of adherence to a healthy diet as a dependent variable (bold values: *p* < 0.05).

**Figure 3 healthcare-12-01006-f003:**
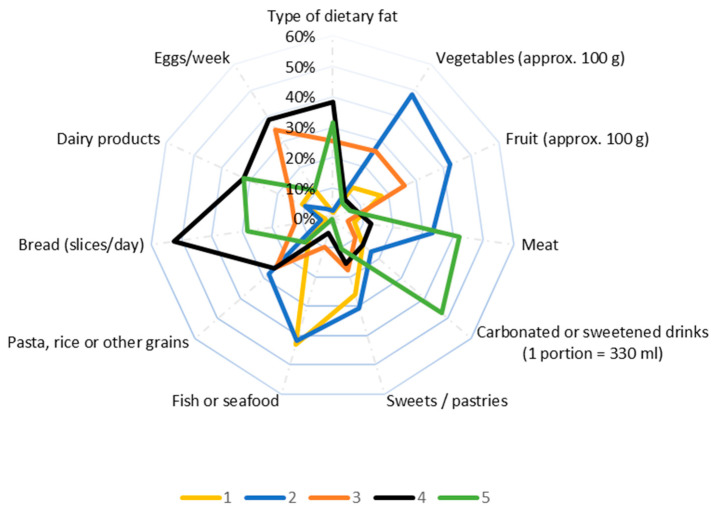
Frequency of consumption of various categories of food products among young people (18–30 years): 1—very rarely or not at all; 2—once a week; 3—twice a week; 4—more than 2 times a week; 5—daily.

**Figure 4 healthcare-12-01006-f004:**
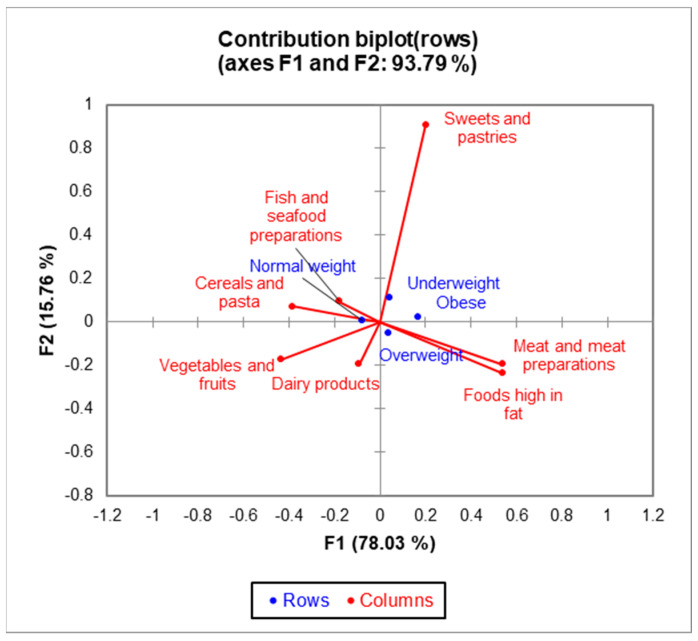
The bi-plot with BMI categories and most consumed food products.

**Figure 5 healthcare-12-01006-f005:**
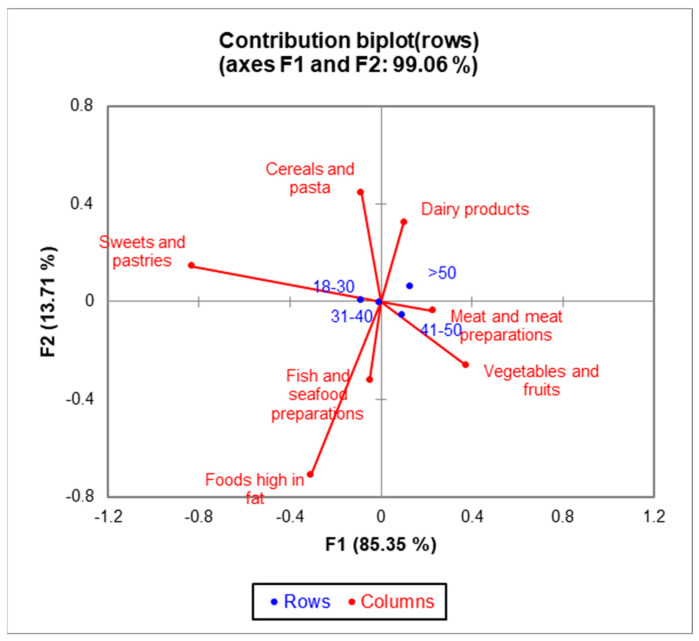
The bi-plot with age categories and most consumed food products.

**Figure 6 healthcare-12-01006-f006:**
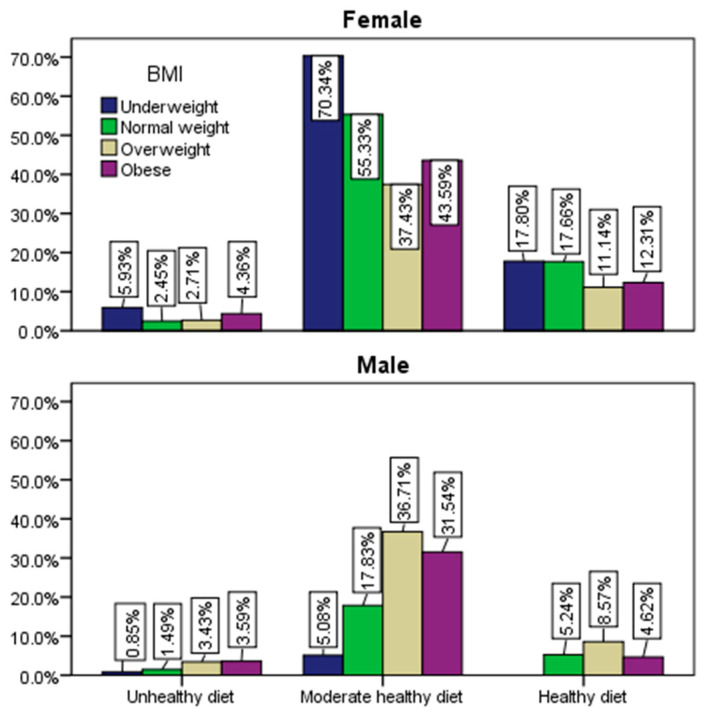
Adherence to a healthy diet by gender and BMI.

**Figure 7 healthcare-12-01006-f007:**
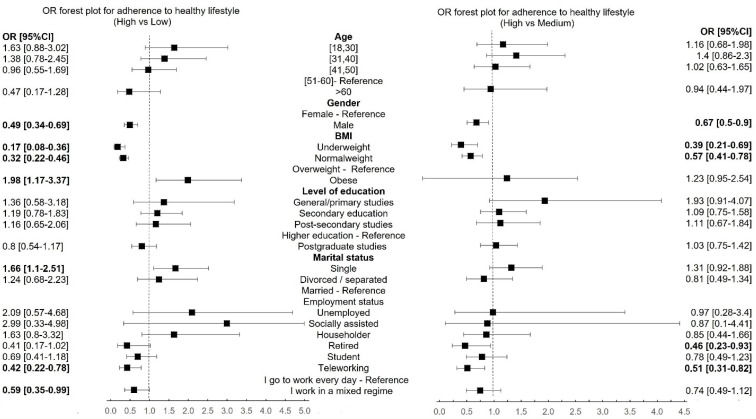
The multinomial logistic regression results with categories of adherence to a healthy lifestyle as a dependent variable (bold values: *p* < 0.05).

**Figure 8 healthcare-12-01006-f008:**
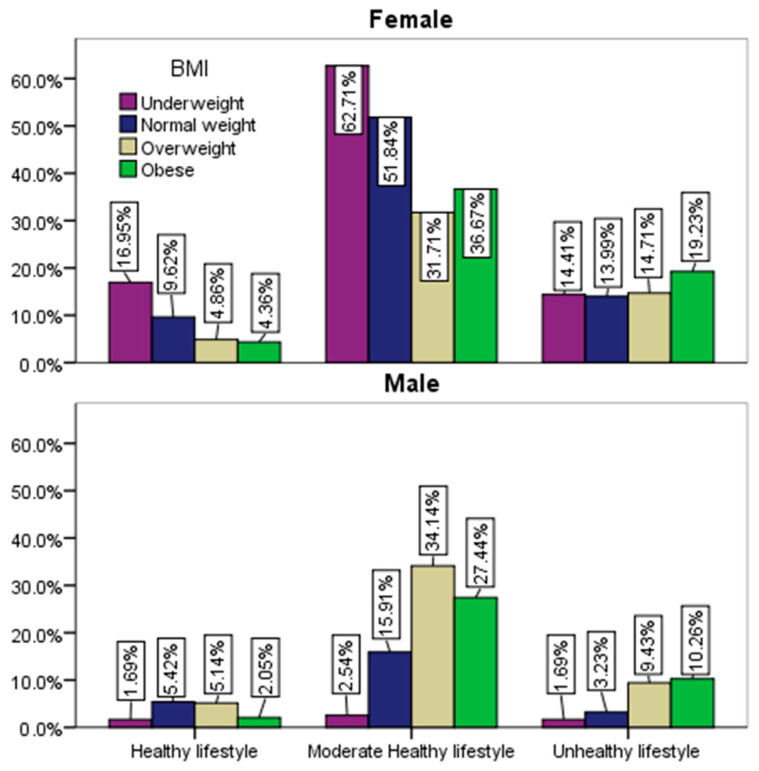
Adherence to a healthy lifestyle by gender and BMI.

**Figure 9 healthcare-12-01006-f009:**
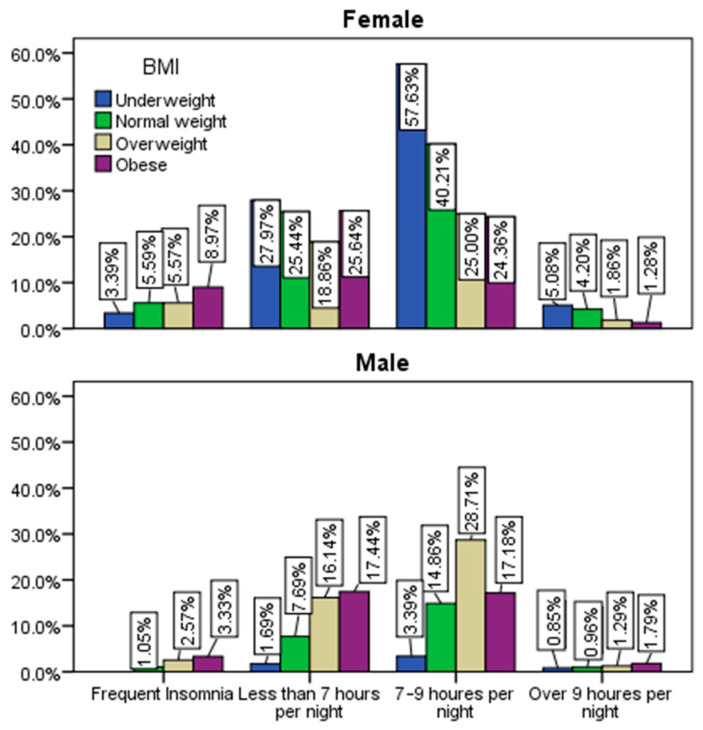
Distribution of gender and sleep duration by BMI.

**Figure 10 healthcare-12-01006-f010:**
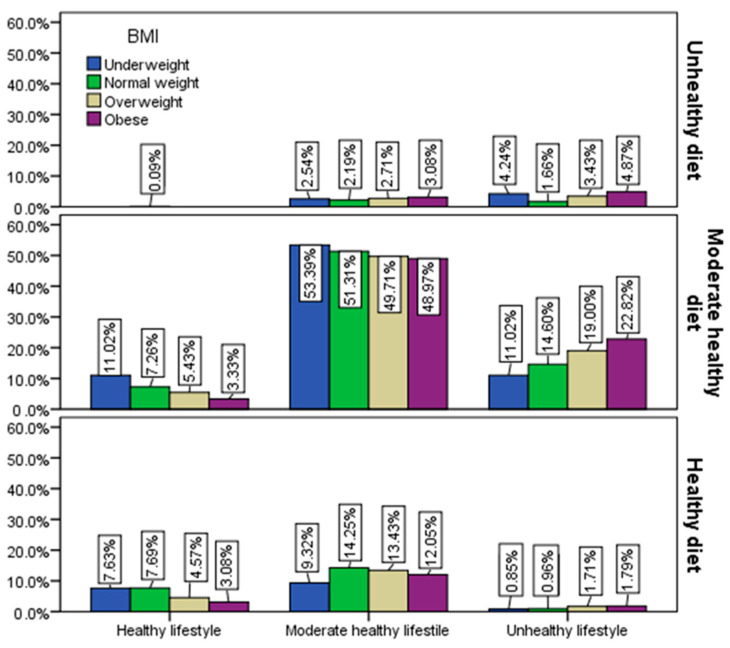
Adherence to a healthy lifestyle and a heathy diet by BMI.

**Figure 11 healthcare-12-01006-f011:**
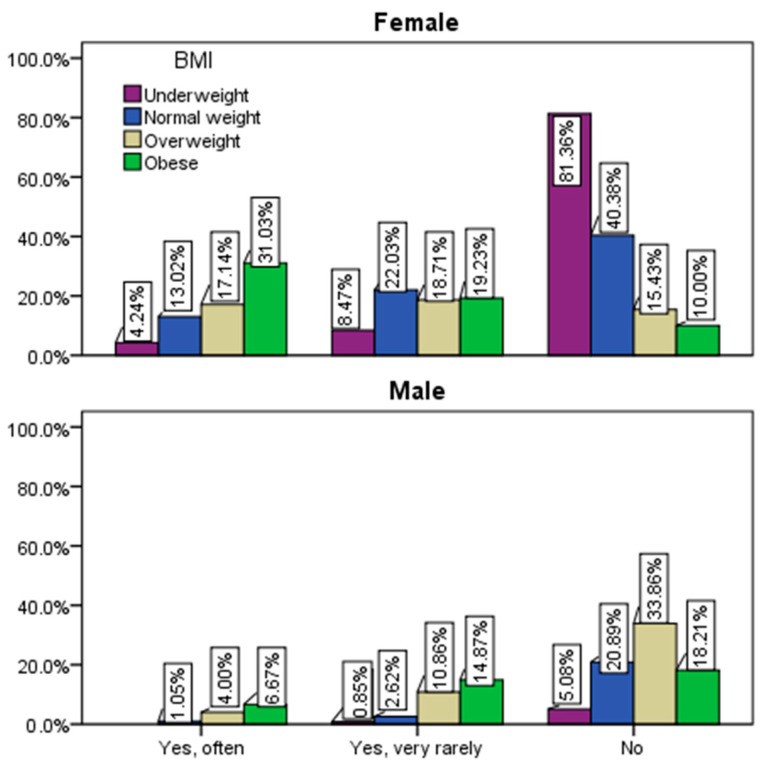
Application of a restrictive diet by gender and BMI.

**Figure 12 healthcare-12-01006-f012:**
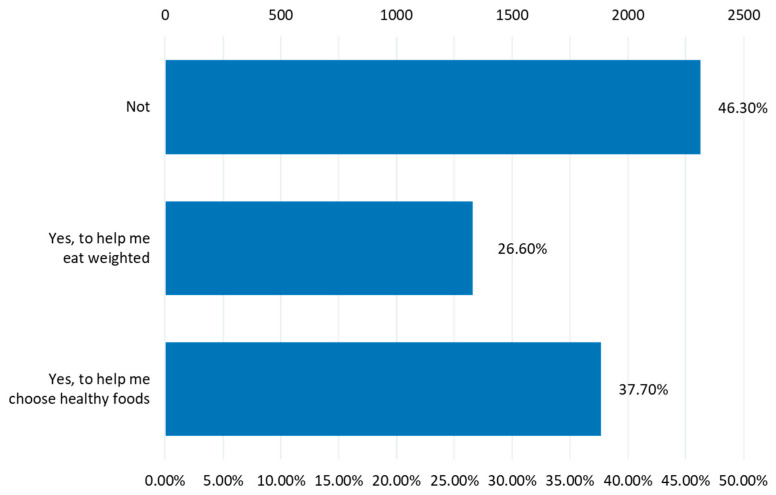
The need for competent advice from nutritionists.

**Figure 13 healthcare-12-01006-f013:**
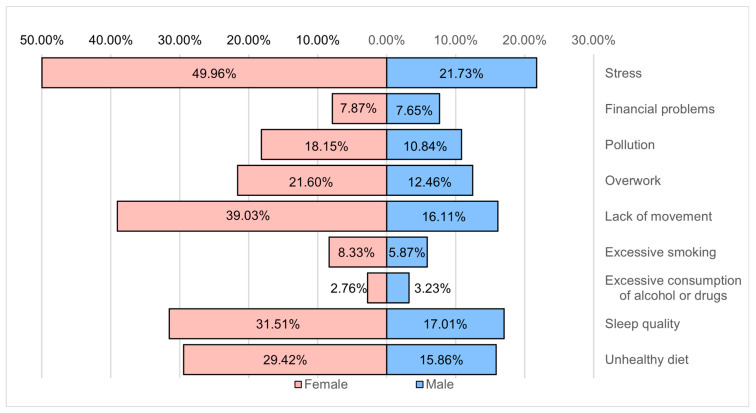
Health risk factors.

**Figure 14 healthcare-12-01006-f014:**
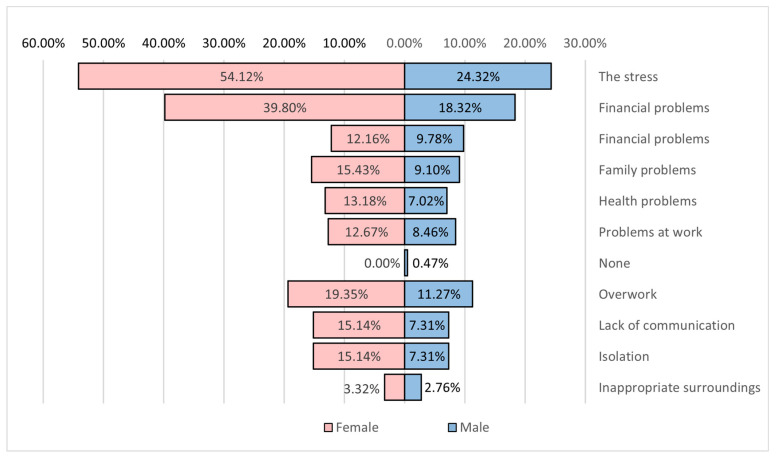
Risk factors for the mental state.

**Figure 15 healthcare-12-01006-f015:**
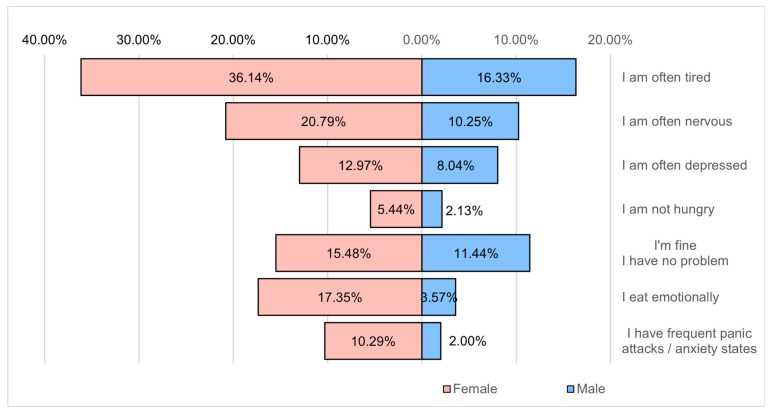
Psycho-emotional problems.

**Figure 16 healthcare-12-01006-f016:**
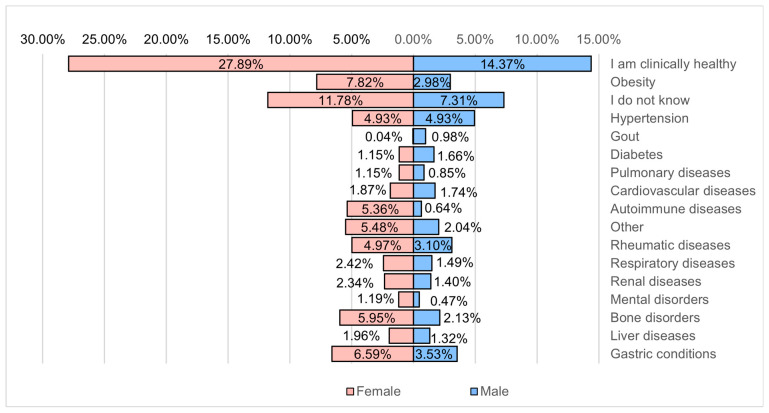
Associated chronic diseases.

**Figure 17 healthcare-12-01006-f017:**
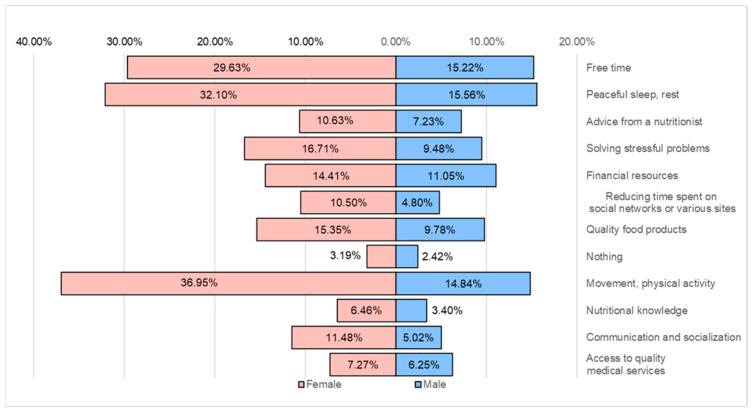
Factors that depreciate lifestyle.

**Figure 18 healthcare-12-01006-f018:**
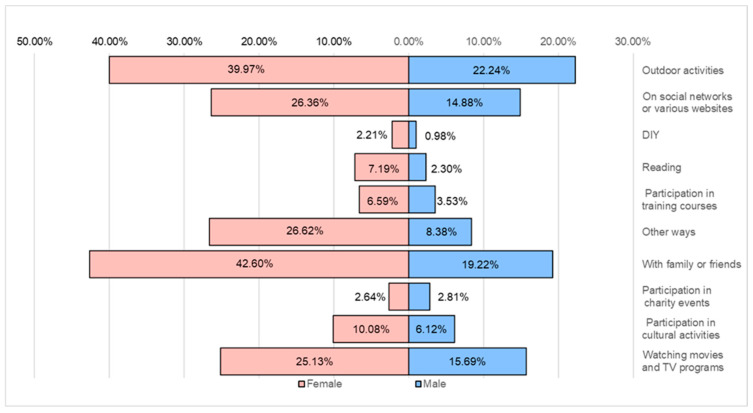
Ways of recreation.

**Table 1 healthcare-12-01006-t001:** Socio-demographic and anthropometric characteristics of respondents.

Characteristics	Total Population—n (%)	Male—n (%)	Female—n (%)
	4704 (100)	1568 (33.3)	3136 (66.7)
**Age (years)**; *p* < 0.0001			
18–30	1756 (37.3)	528 (33.7)	1228 (39.2)
31–40	1136 (24.2)	354 (22.6)	782 (24.9)
41–50	1034 (22)	344 (21.9)	690 (22)
51–60	534 (11.3)	204 (13)	324 (10.5)
>60	246 (5.2)	138 (8.8)	1008 (3.4)
**Residence areas**; *p* = 0.0002			
Urban areas	3648 (77.6)	1144 (72.9)	2504 (79.8)
Rural areas	1056 (22.4)	424 (23.1)	632 (20.2)
**Level of education**; *p* < 0.0001			
General/primary studies	298 (6.3)	154 (9.8)	144 (4.6)
Secondary education	1090 (23.2)	392 (25.0)	698 (22.3)
Postsecondary studies	420 (8.9)	132 (8.4)	288 (9.2)
Higher education	1616 (34.4)	528 (33.7)	1088 (34.7)
Postgraduate studies	1280 (27.2)	362 (23.1)	918 (29.3)
**Marital status**; *p* = 0.0049			
Single	1688 (35.9)	634 (53.7)	1054 (59.6)
Divorced/separated	306 (6.5)	92 (5.9)	214 (6.8)
Married	2710 (57.6)	842 (40.4)	1868 (33.6)
**Employment status**; *p* < 0.0001			
Unemployed	78 (1.7)	34 (2.2)	44 (1.4)
Socially assisted	26 (0.6)	10 (0.6)	16 (0.5)
Householder	312 (6.6)	36 (2.3)	276 (8.8)
Retired	270 (5.7)	140 (8.9)	130 (4.1)
Student	838 (17.8)	210 (13.4)	628 (20)
Teleworking	274 (5.8)	92 (5.9)	182 (5.8)
Going to work every day	2406 (51.1)	848 (54.1)	1558 (49.7)
Mixed working regime	500 (10.6)	198 (12.6)	302 (9.6)
**Body mass index (BMI)**; *p* < 0.0001			
Normal (18.5–24.9)	2288	562 (35.8)	1726 (55.0)
Overweight (25–29.9)	1400	682 (43.5)	718 (22.9)
Underweight (<18.5)	236	14 (0.9)	222 (7.1)
Obese (≥30)	780	310 (19.8)	470 (15.0)

Secondary education—baccalaureate degree; Higher education—bachelor’s degree; Postgraduate studies—master’s degree, residency, doctorate, other specializations; Mixed working regime—teleworking and commuting.

**Table 2 healthcare-12-01006-t002:** Adherence to a healthy diet of age-group, gender, BMI group, Residence areas, Education level, Marital status, and Employment status.

Variable	Healthy Diet		Moderately Healthy Diet		Unhealthy Diet		*p*-Value Chi-Squared Test
	n	%	n	%	n	%	
Total	974	20.7	3476	73.9	234	5.4	
**Age**							
18–30	292	30	1334	38.4	130	51.2	<0.0001
31–40	270	27.7	810	23.3	56	22	
41–50	222	22.8	766	22	44	17.3	
51–60	106	10.9	408	11.7	20	7.9	
>60	84	8.6	158	4.6	4	1.6	
**Gender**							
F	698	71.7	2296	66.1	142	55.9	0.0020
M	276	28.3	1180	33.9	112	44.1	
**BMI**							
Underweight	42	4.3	178	5.1	16	6.2	0.0102
Normal weight	524	53.8	1674	48.1	90	35.4	
Overweight	276	28.3	1038	29.8	86	33.8	
Obese	132	13.6	586	16.8	62	24.4	
**Residence areas**							
Urban	788	80.9	2668	76.8	96	75.6	0.1315
Rural	186	19.1	808	23.2	62	24.4	
**Level of education**							
General/primary studies	36	3.7	248	7.2	14	5.5	0.0087
Secondary education	202	20.7	818	23.5	70	27.5	
Postsecondary studies	84	8.6	300	8.6	36	14.2	
Higher education	344	35.3	1204	34.6	68	26.9	
Postgraduate studies	308	31.6	906	26.1	66	25.9	
**Marital status**							
Single	314	32.2	1254	36.1	120	47.2	0.0394
Divorced/separated	64	6.6	228	6.5	14	5.6	
Married	596	61.2	1994	57.4	120	47.2	
**Employment status**							
Unemployed	8	0.8	64	1.8	6	2.4	0.0038
Socially assisted	10	1	16	0.5	0	0	
Householder	68	6.9	222	6.4	22	8.7	
Retired	84	8.6	182	5.2	4	1.6	
Student	140	14.4	632	18.2	66	25.9	
Teleworking	66	6.7	204	5.9	4	1.6	
Going to work every day	486	49.8	1796	51.7	124	48.8	
Mixed working regime	112	11.5	360	10.3	28	11	

Secondary education—baccalaureate degree; Higher education—bachelor’s degree; Postgraduate studies—master’s degree, residency, doctorate, other specializations; Mixed working regime—teleworking and commuting.

**Table 3 healthcare-12-01006-t003:** Adherence to a healthy lifestyle of age-group, gender, BMI group, Residence areas, Education level, Marital status, and Employment status.

Variable	Healthy Lifestyle		Moderately Healthy Lifestyle		Unhealthy Lifestyle		*p*-Value Chi-Squared Test
	n	%	n	%	n	%	
Total	578	12.28	3126	66.45	1000	21.25	
**Age**							
18–30	208	35.99	1118	35.76	430	43.00	0.0038
31–40	120	20.76	790	25.27	226	22.60	
41–50	134	23.18	698	22.33	200	20.00	
51–60	66	11.42	350	11.20	118	11.80	
>60	50	8.65	170	5.44	26	2.60	
**Gender**							
F	362	62.63	2064	66.03	710	71.00	0.0363
M	216	37.37	1062	33.97	290	29.00	
**BMI**							
Underweight	44	7.61	154	4.93	38	3.80	0.0001
Normal weight	344	59.52	1550	49.58	394	39.40	
Overweight	140	24.22	922	29.49	338	33.80	
Obese	50	8.65	500	15.99	230	23.00	
**Residence areas**							
Urban areas	484	83.74	2384	76.26	80	78.00	0.0193
Rural areas	94	16.26	742	23.74	220	22.00	
**Level of education**							
General/primary studies	20	3.46	218	6.97	60	6.00	0.0167
Secondary education	124	21.45	692	22.14	274	27.40	
Postsecondary studies	46	7.96	268	8.77	100	10.00	
Higher education	212	36.68	1056	33.78	348	34.80	
Postgraduate studies	176	30.45	886	28.34	218	21.80	
**Marital status**							
Single	190	32.87	1104	35.32	394	39.40	0.1682
Divorced/separated	46	7.96	188	6.01	72	7.20	
Married	342	59.17	1834	58.67	534	53.40	
**Employment status**							
Unemployed	6	1.04	42	1.34	10	3.00	<0.0001
Socially assisted	2	0.35	14	0.45	10	1.00	
Householder	24	4.15	182	5.82	106	10.60	
Retired	58	10.03	178	5.69	34	3.40	
Student	12	17.65	542	17.34	194	19.40	
Teleworking	54	9.34	178	5.57	46	4.60	
Going to work every day	258	44.64	1648	52.72	500	50.00	
Mixed working regime	74	12.48	346	10.7	80	8.00	

Secondary education—baccalaureate degree; Higher education—bachelor’s degree; Postgraduate studies—master’s degree, residency, doctorate, other specializations; Mixed working regime—teleworking and commuting.

## Data Availability

Data are contained within the article and Appendix A.

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
