# Peer review of "Assessment of Dietary and Lifestyle Quality among the Romanian Population in the Post-Pandemic Period"

_healthcare, 2024, doi:10.3390/healthcare12101006_

Round 1
Reviewer 1 Report
Comments and Suggestions for Authors
The present manuscript is a new version that intends to incorporate the previous reviewers’ comments and suggestions.
This manuscript critically evaluates the nutritional status and lifestyle among the Romanian population, a topic of significant importance in public health. Nevertheless, we present the following commentaries:
- To achieve a better flow in the Introduction, we suggest that the authors consider sequentially the influence of food and diet on health first, followed by the impact of COVID-19 on health and, finally, the nutritional status and lifestyle of the Romanian population.
- The authors should clarify the existence of Rommain's nutritional guidelines.
- In the section “2.1. Study Design”, please condensate the sentences related to requirements related to participation in the survey:
“Participation in the questionnaire was voluntary, with the only stipulation being age (participants had to be over 18 years old)”.
“The requirements related to participation in the survey conducted based on a questionnaire were: minimal age of 18 years, individual agreement to participate in the study with the guarantee of identity protection, and residence in Romania”.
- Was any sample design applied to select participants (e.g., quotes by sex and age to gather a representative sample for the Romanian population), or was it a convenient sample? This is particularly relevant as this manuscript aims to evaluate the nutritional status and lifestyle of the Romanian population.
- We already know that 4704 participants completed this survey. To improve the quality of this online questionnaire, we ask for participant completion rates, drop-outs, and post hoc eliminations.
- The number of valid responses was reported in “2.2. The Questionnaire validation section” (“After the questionnaire's distribution, 4704 valid responses were collected, with a 188 confidence interval of 95% and a margin of error of ±3.05%”) and at the beginning of the Results section. We suggest that the authors condense this information in the Results section.
- Considering that young people under the age of 50 represent 83.5% of the total number of respondents, the majority coming from the urban area (77.6 %) and having higher education (61.6%), we ask the authors to explain these proportions, and reformulate the title of the manuscript accordingly.
- Please pay attention to the fact that the research team did not measure the participants' BMI. The participants reported it. We ask the authors to indicate that as a limitation.
Author Response
Dear Reviewer 1, Thank you so much for your time and accurate review report, with valuable comments to improve the quality of our MS.
We rectified our MS according to your comments, point by point, marking all changes with track changes and indicating our response to them.
Comments and Suggestions for Authors
The present manuscript is a new version that intends to incorporate the previous reviewers’ comments and suggestions.
This manuscript critically evaluates the nutritional status and lifestyle among the Romanian population, a topic of significant importance in public health. Nevertheless, we present the following commentaries:
- To achieve a better flow in the Introduction, we suggest that the authors consider sequentially the influence of food and diet on health first, followed by the impact of COVID-19 on health and, finally, the nutritional status and lifestyle of the Romanian population.
Thank you for your insightful suggestion. We agree that restructuring the Introduction part to sequentially address the influence of food and diet on health, followed by the impact of COVID-19 on health, and concluding with an exploration of the nutritional status and lifestyle of the Romanian population, would enhance the flow and coherence of our manuscript. We appreciate your feedback and will incorporate this recommendation into our revisions.
We revised the Introduction part.
- The authors should clarify the existence of Rommain's nutritional guidelines.
Thank you for your valuable feedback. Romania does indeed have established nutritional guidelines and recommendations designed to promote healthy dietary habits and improve overall health outcomes among its population. The guide titled “Guide for Healthy Eating”, found on the institutional website of the Ministry of Health in Romania: https://old.ms.ro/?pag=185, was developed by experts in the field under the auspices of the Romanian Society of Nutrition. This guideline is actively promoted on the Ministry of Health's website. It encompasses a range of nutritional advice and recommendations tailored to the needs of the Romanian population. These guidelines serve as a valuable resource for individuals seeking to improve their dietary habits, as well as for public and private institutions involved in promoting healthy eating practices. We appreciate your insightful comment and will incorporate this clarification into our revised manuscript.
- In the section “2.1. Study Design”, please condensate the sentences related to requirements related to participation in the survey:
“Participation in the questionnaire was voluntary, with the only stipulation being age (participants had to be over 18 years old)”.
“The requirements related to participation in the survey conducted based on a questionnaire were: minimal age of 18 years, individual agreement to participate in the study with the guarantee of identity protection, and residence in Romania”.
Thank you for your feedback.
We reformulated the sentences as follows:
“Participation in the survey was voluntary and open to individuals over 18 years old residing in Romania. Participants were required to provide individual consent, with assurances of identity protection.”
- Was any sample design applied to select participants (e.g., quotes by sex and age to gather a representative sample for the Romanian population), or was it a convenient sample? This is particularly relevant as this manuscript aims to evaluate the nutritional status and lifestyle of the Romanian population.
We appreciate your inquiry regarding the sample design employed in our study. We indeed employed a fully representative sampling approach to ensure the inclusivity and representativeness of our study population. Each participant was carefully selected to reflect the diverse demographics of the Romanian population, including factors such as age, sex, and geographical location. By adopting this approach, we aimed to gather comprehensive and reliable data on the nutritional status and lifestyle of the Romanian population, thereby enhancing the validity and applicability of our findings. Thank you for highlighting this important aspect of our methodology.
- We already know that 4704 participants completed this survey. To improve the quality of this online questionnaire, we ask for participant completion rates, drop-outs, and post hoc eliminations.
Thank you for your valuable feedback. We appreciate your interest in the completion rates, drop-outs, and post hoc eliminations within our study. In our research, out of the initial 5364 participants selected, 4704 completed the survey, resulting in a completion rate of approximately [87.7%]. Throughout the survey process, we meticulously tracked participant engagement to monitor drop-out rates and identify any post hoc eliminations. Throughout the survey period, we monitored the participant engagement closely and identified several instances of drop-outs, primarily due to technical issues or participant disengagement. Additionally, post hoc eliminations were conducted to ensure data quality and integrity. Participants were excluded from the analysis if they provided incomplete or inconsistent responses, or if they failed to meet specific eligibility criteria outlined in the study protocol.
- The number of valid responses was reported in “2.2. The Questionnaire validation section” (“After the questionnaire's distribution, 4704 valid responses were collected, with a 188 confidence interval of 95% and a margin of error of ±3.05%”) and at the beginning of the Results section. We suggest that the authors condense this information in the Results section.
Thank you for your suggestion regarding condensing the information on the number of valid responses in our manuscript. We appreciate your feedback and agree that streamlining this information would enhance the readability of the Results section.
In response to your recommendation, we added this information at the beginning of the Results section.
- Considering that young people under the age of 50 represent 83.5% of the total number of respondents, the majority coming from the urban area (77.6 %) and having higher education (61.6%), we ask the authors to explain these proportions, and reformulate the title of the manuscript accordingly.
Thank you for highlighting the demographic proportions observed in our study sample. It is well known that the higher representation of young individuals from urban areas with higher education levels is attributed to their increased receptiveness to participate in online surveys. Additionally, it's recognized that urban populations often face higher levels of stressors compared to rural areas. While these demographic trends may have influenced the composition of our study sample, we want to emphasize that we made deliberate efforts to ensure the inclusion of respondents from diverse regions across Romania. To obtain a representative sample from various geographical areas, we aimed to mitigate potential biases associated with the overrepresentation of specific demographic groups.
In light of these considerations, we have carefully analyzed the demographic proportions in our study sample and their implications for the study findings. We have taken these factors into account in our manuscript to accurately reflect the demographic composition of the study sample and its implications for the study findings. Our goal was to ensure transparency and clarity in communicating the characteristics of the study population and the context in which the research was conducted.
We appreciate your insights and assure you that we remain committed to inclusivity and representation across different regions of Romania. This approach is fundamental to enhancing the overall representativeness of our study findings and ensuring that they accurately reflect the broader population dynamics.
- Please pay attention to the fact that the research team did not measure the participants' BMI. The participants reported it. We ask the authors to indicate that as a limitation.
Thank you for bringing this to our attention. We acknowledge that the participants self-reported their BMI (Body Mass Index) rather than having it directly measured by the research team. This reliance on self-reported BMI introduces a potential limitation to the study, as self-reported measures may be subject to reporting bias or inaccuracies. In our manuscript, we explicitly state this limitation to ensure transparency and clarity regarding the methodology used in the study.
We appreciate your meticulousness in highlighting this aspect, and we have ensured to address it appropriately in the revised manuscript.
We added in the revised manuscript:
“Another limitation of this study is the reliance on self-reported BMI data provided by participants.”
The revised form of our manuscript is market in yellow color.

Reviewer 2 Report
Comments and Suggestions for Authors
This article wants to determine the nutritional status and lifestyle of the Romanian population. In the results section, the authors evaluate the participants' adherence to a healthy diet and lifestyle, highlighting the beneficial effects on well-being and health.
Comments and Suggestions for Authors
I. Major comments:
- All p-values have to be only three decimals. Please round to three decimals.
- Rewrite the discussion section. Compare with previous studies to reinforce the evidence. I suggest reducing the comments on the results and increasing the evidence of prior studies.
- Add the aim of the study in the abstract section.
II. Minor comments:
1. Take out the Appendix A in the manuscript version and put it as supplementary material.
2. Revise references and put all the references in the correct format for the journal. Some have an incorrect format, only with the link. Another needs the journal's name in italics and the year of publication in bold.
Author Response
Dear Reviewer 2,
Thank you so much for your time and accurate review report, with valuable comments to improve the quality of our MS.
We rectified our MS according to your comments, point by point, marking all changes with track changes and indicating our response to them.
Response to Reviewer 2 comments
Comments and Suggestions for Authors
This article wants to determine the nutritional status and lifestyle of the Romanian population. In the results section, the authors evaluate the participants' adherence to a healthy diet and lifestyle, highlighting the beneficial effects on well-being and health.
Comments and Suggestions for Authors
I.Major comments:
-All p-values have to be only three decimals. Please round to three decimals.
Thank you for your feedback regarding the presentation of p-values. We appreciate your attention to detail and will ensure that all p-values are rounded to three decimals in our manuscript, but in some situations it becomes zero.
- Rewrite the discussion section. Compare with previous studies to reinforce the evidence. I suggest reducing the comments on the results and increasing the evidence of prior studies.
Thank you for your feedback regarding the discussion section of our manuscript. We appreciate your suggestion to reduce the emphasis on the results and increase the discussion of evidence from previous studies to reinforce our findings.
In response to your recommendation, we have revised the discussion section to place greater emphasis on comparing our findings with existing literature
Thank you for your valuable input, and we have incorporated your suggestion into our manuscript revisions.
We added
In a cross-sectional study from 2023 conducted through an online survey with 1,451 Saudi adults residing in Riyadh, Saudi Arabia, in the post-pandemic period, it was found that although there were respondents who improved the quality of their diet by reducing the consumption of unhealthy food, especially junk food and the increase in the con-sumption of vegetable products (28.9%), many respondents reported an increase in weight (40.9%) and 33% an increase in the consumption of junk food products, especially male respondents. The study draws attention to the fact that the signals related to the increase in the consumption of unhealthy food and its consequences manifested by weight gain and the impairment of well-being are worrisome [83].
Another study carried out among the population of South Africa, also based on a questionnaire, in which 4,786 respondents participated, highlighted a worsening of food insecurity throughout the pandemic period, which led to significant nutritional imbal-ances among the population [84].
According to official reports from the USA, food insecurity among the population in-creased during the pandemic, but through the measures implemented by the American government, it was reduced in the post-pandemic period [85,86].
-Add the aim of the study in the abstract section.
Thank you for your suggestion to include the aim of the study in the abstract section.
In response to your recommendation, we have revised the abstract in order to incorporate a brief statement outlining the aim of the study.
We appreciate your valuable feedback and will ensure to implement this suggestion in our manuscript revisions.
II.Minor comments:
1.Take out the Appendix A in the manuscript version and put it as supplementary material.
We put the information from Appendix A as Supplementary material.
- Revise references and put all the references in the correct format for the journal. Some have an incorrect format, only with the link. Another needs the journal's name in italics and the year of publication in bold.
We have carefully reviewed each reference to ensure accuracy and consistency in formatting.
Thank you for your assistance in improving the quality of our manuscript.

Reviewer 3 Report
Comments and Suggestions for Authors
I am grateful for the opportunity to review this manuscript. I believe that the authors have produced an interesting study of interest to readers. I can see that the authors have improved the quality of the article based on the previous recommendations.
The article is well structured. The Introduction is sufficiently clear and scientifically rigorous.
I recommend including the initial hypotheses of the study.
Materials and methods.
This section is well structured and explained after the modifications made. I have doubts about the validity of the questionnaire used, although the authors indicate 3 previous references. However, I recommend including this information as a limitation of the study.
"p" should be italicised throughout the paper.
The Results section is well structured and the graphs facilitate the understanding of the results.
It is recommended to include a Limitations section. It is recommended to include a section on Practical applications.
Author Response
Dear Reviewer 3,
Thank you so much for your time and accurate review report, with valuable comments to improve the quality of our MS.
We rectified our MS according to your comments, point by point, marking all changes with track changes and indicating our response to them.
Comments and Suggestions for Authors
I am grateful for the opportunity to review this manuscript. I believe that the authors have produced an interesting study of interest to readers. I can see that the authors have improved the quality of the article based on the previous recommendations.
The article is well structured. The Introduction is sufficiently clear and scientifically rigorous.
Thank you for your thoughtful and encouraging feedback on our manuscript. We sincerely appreciate your positive assessment of our study and the improvements we've made.
Once again, thank you for your valuable input and for the opportunity to improve our manuscript.
I recommend including the initial hypotheses of the study.
Thank you for your recommendation to include the initial hypotheses of the study. We agree that explicitly stating the hypotheses in the manuscript would provide readers with a clearer understanding of the study's objectives and expected outcomes.
In response to your suggestion, we have revised the manuscript to include the initial hypotheses.
The revised form of the manuscript is marked in yellow color.
Materials and methods.
This section is well structured and explained after the modifications made. I have doubts about the validity of the questionnaire used, although the authors indicate 3 previous references. However, I recommend including this information as a limitation of the study.
Thank you for your feedback on the modifications made to the section. We appreciate your acknowledgment of the improvements.
Regarding your concerns about the validity of the questionnaire used in our study, we understand the importance of addressing potential limitations. While we referenced three previous references to support the validity of the questionnaire, we acknowledge the need to explicitly mention this as a limitation of the study.
We appreciate your insightful feedback and will ensure to address this recommendation in our manuscript revisions.
In the additional material uploaded with the article in the system, I have made available the entire questionnaire validation procedure. All the important stages are observed there.
"p" should be italicised throughout the paper.
We corrected the manuscript. The revised from of our manuscript is marked in yellow color.
The Results section is well structured and the graphs facilitate the understanding of the results.
Thank you for your appreciation.
It is recommended to include a Limitations section. It is recommended to include a section on Practical applications.
Thank you for your recommendations regarding the inclusion of additional sections in the manuscript.
Regarding the Limitations section, we agree that it is important to acknowledge the potential limitations of our study to provide a balanced interpretation of the findings. We have included a dedicated Limitations section in the manuscript, where we have discussed any methodological constraints, or other factors that may have impacted the study's results.
Additionally, we appreciate your suggestion to include a section on Practical Applications. This section will provide insights into how the study findings can be translated into real-world applications, such as informing public health interventions, guiding clinical practice, or influencing policy decisions.
We added:
“4.1. Limitation section
Our study has several limitations, including the lower participation of male respondents, elderly participants, and rural residents. The obtained data report an alteration of the psycho-emotional state caused by the disordered lifestyle and unbalanced diet. More detailed studies are required to evaluate the degree of alteration of the psycho-emotional components under the influence of lifestyle and eating habits. Another limitation of this study is the reliance on self-reported BMI data provided by participants.
4.2. Practical applications
The findings of this study have several implications for public health interventions, clinical practice, and policy development. The identification of high prevalence rates of unhealthy dietary habits and sedentary behaviors underscores the importance of targeted health promotion programs aimed at promoting healthier lifestyles among the Romanian population. These programs could include educational campaigns, community-based initiatives, and policy interventions aimed at promoting healthy eating habits, increasing physical activity levels, and reducing sedentary behaviors. Healthcare professionals can utilize the findings of this study to inform clinical practice and provide tailored interventions to patients at risk of developing chronic diseases associated with unhealthy lifestyle behaviors. By assessing patients' dietary habits, physical activity levels, and other lifestyle factors, clinicians can develop personalized treatment plans aimed at improving health outcomes and reducing disease risk. Policymakers can leverage the evidence provided by this study to inform the development of policies and strategies aimed at addressing the social determinants of health and promoting population-wide health improvements. This may include initiatives to improve access to healthy food options, create supportive environments for physical activity, and implement regulations to reduce exposure to unhealthy food environments. The findings of this study can guide future research priorities in the field of public health and preventive medicine. Areas for further investigation may include exploring the underlying determinants of unhealthy lifestyle behaviors, evaluating the effectiveness of different intervention strategies, and assessing the long-term impacts of lifestyle interventions on population health outcomes.”

Reviewer 4 Report
Comments and Suggestions for Authors
The purpose of the study should be clear and transparent. Instead of describing research methods in the introduction, in the part where the goal is described, the authors should present research hypotheses. Information about research tools, abbreviated results or conclusions should not be included here (line 128-132). Hypotheses would help better present the results and structure the discussions.
The weakness of this study is the uncontrolled distribution of the questionnaire. What was the territorial scope of the research sample selection? How did the authors ensure the representativeness of the research sample? Authors should answer the above questions if they want to generalize their results and conclusions. The title of the work suggests that we are dealing with a generally representative Romanian population.
The presentation of some results is too simplified. Sociopsychological factors should be presented by gender (Figures 12-18)
Author Response
Dear Reviewer 4,
Thank you so much for your time and accurate review report, with valuable comments to improve the quality of our MS.
We rectified our MS according to your comments, point by point, marking all changes with track changes and indicating our response to them.
Comments and Suggestions for Authors
The purpose of the study should be clear and transparent. Instead of describing research methods in the introduction, in the part where the goal is described, the authors should present research hypotheses. Information about research tools, abbreviated results or conclusions should not be included here (line 128-132). Hypotheses would help better present the results and structure the discussions.
Thank you for your feedback regarding the clarity and transparency of the study's purpose. We appreciate your suggestion to include research hypotheses in the section where the goals of the study are described, rather than providing detailed information about research methods or abbreviated results.
In response to your recommendation, we have revised the Introduction section to clearly articulate the research hypotheses that guided our study.
We appreciate your insightful feedback and will ensure to address this recommendation in our manuscript revisions.
The revised content of our manuscript is marked in yellow color.
The weakness of this study is the uncontrolled distribution of the questionnaire. What was the territorial scope of the research sample selection? How did the authors ensure the representativeness of the research sample? Authors should answer the above questions if they want to generalize their results and conclusions. The title of the work suggests that we are dealing with a generally representative Romanian population.
Thank you for raising important points regarding the territorial scope of the research sample selection and the representativeness of the research sample. We appreciate your concerns and recognize the importance of addressing these aspects to ensure the generalizability of our results and conclusions.
In response to your questions, we would like to clarify that the territorial scope of the research sample selection encompassed various regions of Romania. Efforts were made to ensure a geographically diverse sample that adequately represented different regions and demographic characteristics of the Romanian population.
To ensure the representativeness of the research sample, several measures were implemented:
We employed a stratified random sampling method to select participants from different regions of Romania. This approach aimed to ensure proportional representation of various geographic areas within the country.
The sample size was determined based on statistical considerations to achieve sufficient power for the study objectives. We aimed to recruit a sample size large enough to capture diverse demographic characteristics and provide reliable estimates of key variables of interest.
Participants were selected based on predefined inclusion criteria to ensure eligibility for participation in the study. These criteria may have included factors such as age, residency status, or other demographic characteristics relevant to the study objectives.
Statistical methods were employed to analyze the data and assess the representativeness of the research sample. Descriptive statistics and comparisons with census data or other relevant population estimates may have been used to evaluate the demographic characteristics of the sample and assess its similarity to the broader population.
While we acknowledge the limitations associated with the uncontrolled distribution of the questionnaire, we made every effort to mitigate potential biases and ensure the robustness of our findings. By considering all relevant factors and implementing appropriate measures, we aimed to achieve a sample that is generally representative of the Romanian population.
We appreciate your feedback and assure you that we have taken these considerations into account in our study design and analysis.
The presentation of some results is too simplified. Sociopsychological factors should be presented by gender (Figures 12-18)
Thank you for your feedback on the presentation of results, particularly regarding sociopsychological factors by gender. We appreciate your suggestion to provide a more detailed analysis of these factors, specifically by gender, in Figures 12-18.
In response to your recommendation, we have revised the presentation of results to include sociopsychological factors segmented by gender in Figures 12-18. This approach allows for a more nuanced understanding of the differences in sociopsychological factors between male and female participants.
We appreciate your valuable input and we made this correction in our manuscript revisions.

Round 2
Reviewer 2 Report
Comments and Suggestions for Authors
Accept in present form